# Mothers' Perceptions of the Phenomenon of Bullying among Young Children in South Korea

**Hyun-jung Ju and Seung-ha Lee ***

Department of Early Childhood Education, Chung-Ang University, 84 Heukseok-ro, Dongjak-gu, Seoul 712-749, Korea; jhj834@cau.ac.kr

**\*** Correspondence: seungha94@cau.ac.kr; Tel.: +82-2-820-5882

**Abstract:** This study aimed to investigate mothers' different perspectives on bullying in early childhood. Twelve mothers having children under eight years old were interviewed in South Korea. All the interviews were transcribed in Korean and analyzed using Nvivo. The constant comparison method was used to analyze the data. The results showed six themes consisting of categories: (1) concept of bullying (2) difficulty in defining bullying in early childhood, (3) difficulty in telling other mothers about bullying, (4) children who do not reveal their experiences, (5) ways to be aware of bullying, and (6) mothers' concern. Categories were sometimes divided into subcategories. Findings showed that mothers seemed to view bullying differently, and that relationships among them contributed to differences in their perspectives on bullying. Mothers' relationships also interacted with children's relationships. Children were unlikely to tell their victimization experiences, due to certain reasons. These findings can contribute to understanding the nature of bullying in early childhood, increasing the social awareness of bullying among young children, and emphasizing the need for intervention/prevention programs.

**Keywords:** bullying; young children; *wang-ta*; mothers; aggression

## 1. Introduction

Bullying is a pervasive problem that can occur at any point from early childhood to adulthood. Bullying is usually defined as a subtype of aggressive behaviors that are repeated over time, toward an individual who is unable to defend oneself (i.e., imbalance of power) (Olweus 1993). This aggression could be physical (e.g., hitting), verbal (e.g., name calling), relational (e.g., exclusion), or cyber (e.g., sending nasty messages through electronic media). The characteristics of bullying in early childhood (Lee et al. 2016; Monks et al. 2003; Monks and Smith 2006) differ from those in middle childhood and adolescence. Young children are less likely than older groups to perceive imbalances of power, and repetition of actions; which may reflect that bullying is less targeted toward one child (Monks and Smith 2006).

Whether the three criteria that are used to define bullying—intentional harm-doing, repetition, and imbalance of power—should be included in the context of young children is a controversial topic. Young children begin engaging in goal-directed behavior, and start perceiving others' intentional behaviors at around the age of four. This is also the age at which they are able to recognize power imbalances between people (Williams et al. 2016). However, due to the instability of the victim's role among young children (Monks et al. 2002), it is unclear whether the repetition of behavior is a necessary aspect of the definition of bullying (Vlachou et al. 2011; Williams et al. 2016). Because of these reasons, some researchers use the terms "precursory bullying" (Levine and Tamburrino 2014) or "unjustified aggression" (Monks et al. 2002).

Despite the controversy regarding the definition of bullying in early childhood, studies have identified aggressive behaviors in early childhood that can be categorized as bullying. Using Olweus' definition of bullying, Kirves and Sajaniemi (2012) reported that 7.1% of three to six-year-olds in their study were bullies, 3.3% were victims, and 2.2% were bully-victims. Additionally, participant roles such as bully, victim, or defenders, were clear, even among three-year-old children and among children under seven years old (Lee et al. 2016; Monks et al. 2002; Perren and Alsaker 2006; Repo and Sajaniemi 2015).

Identifying bullying in the early years is vital to prevent children's social and emotional problems. Experiences of bullying in early childhood were related to difficulties in peer relationships, and they resulted in adjustment problems, as well as internalizing and externalizing behaviors. Young children who bullied others showed low social competence, and were less likely to positively interact with peers (Camodeca et al. 2015). Additionally, their bullying behaviors were often related to victimization (Lee et al. 2016; Perren and Alsaker 2006). Longitudinal studies showed the causality more clearly. Relationally victimized children in kindergarten exhibited high levels of adjustment problems; greater loneliness, frequent school avoidance, and low level of school liking. They were also found to be negatively correlated to school academic achievement (Kochenderfer and Ladd 1996). Arseneault et al. (2006) conducted a cohort study among 2232 children over a five-year period. In their study, earlier pure victims exhibited more internalizing behavior and unhappiness compared to the control group two years later. Earlier bully–victims exhibited more internalizing and externalizing behaviors, compared to the control group and pure victims, two years later. Further, victimization experiences in the first, third, and fifth grades in elementary school predicted a decrease in popularity, an increase in aggressive behaviors, inattention, delinquency, and symptoms of anxiety and depression for five years (Hanish and Guerra 2002).

Parental perceptions of bullying have been explored in several studies. A common finding is that parents are unlikely to be aware of bullying or victimization experiences by their children (Mishna et al. 2006). Generally, children do not tell their parents about being bullied (Mishna 2004), because they think that it would aggravate the situation, and result in the termination of friendships. Furthermore, even when parents are aware of the existence of a problem, they find it difficult to determine whether the situation involves bullying; they are unsure of whether the incident is a common conflict among peers; or whether the problem is the fault of their child or of other children (Mishna 2004).

Identifying bullying is important, as it influences how parents manage the situation (Sawyer et al. 2011). If adults do not act on children's reports of bullying or victimization experiences, children may feel that there is no way to obtain help to escape bullying, exposing them to more bullying and increasing their fear (Clarke and Kiselica 1997). Furthermore, some adults are not even aware that indirect relational aggression or social exclusion are regarded as bullying (Harcourt et al. 2014; Sawyer et al. 2011). Therefore, adults' accurate awareness of what constitutes bullying is necessary. However, among young children, it can be difficult to distinguish bullying from the general negative interactions that can frequently occur among friends, posing a further hindrance for adults in addressing bullying among children (Purcell 2012; Sawyer et al. 2011).

Considering the characteristics described above, defining and recognizing bullying is more difficult in early childhood compared to other age ranges. The few existing studies on the perceptions of young children's bullying are based on the Western world, although the features of bullying may vary across different cultures (Smith et al. 2002).

## 1.1. Bullying in South Korea

The phenomenon of bullying differs across cultures, and the differences may be particularly stark between Western and Eastern countries, which are often compared in terms of collectivistic or individualistic perspectives. South Korea is a collectivistic culture in which people are interdependent with their in-group members, and where harmony among them is prioritized (Hofstede and Hofstede 2005). In South Korea, there are several terms that indicate phenomena similar to bullying: *gipdan-ttadolim* (group isolation), *gipdan-goerophim* (group bullying or group teasing), *hakkyo-pokryuk*

(school violence), and *wang-ta* (social exclusion or excluded/victimized person). These terms have been used interchangeably, although there are some differences between them (Lee et al. 2012). For instance, the term *wang-ta* indicates social exclusion: literally, a socially excluded person. The term *wang-ta* usually include other aggressive behavior (physical or verbal), as well as social exclusion (Lee et al. 2011, 2012). During the late 1990s to the early 2000s, the terms *gipdan-ttadolim* (group isolation) or *gipdan-goerophim* (group bullying or group teasing) were relatively frequently used, whereas the term *hakkyo-pokryuk* has been widely used more recently. The term *hakkyo-pokryuk* includes a wide range of aggressive behavior that may occur among school pupils, and is often used by the formal system. The term *wang-ta* has been more popularized among school pupils (Lee et al. 2012). Koreans do not interpret each term in the same way, although these terms have some similarities. There is no academic agreement on the term that is most likely to correspond to the term bullying in English. Furthermore, there are no studies on the most appropriate Korean term to represent bullying in early childhood. Some studies investigating bullying in early childhood borrow the term *hakkyo-pokryuk* (e.g., Kwak and Kim 2016), because there is no academic agreement or discussion on the term issue. Bullying in early childhood in South Korea has been rarely researched: only a few studies have investigated its prevalence. In Song and Lee (2014) study, 53% of kindergarten teachers observed *ttadolim* among children over the course of a year, and most of them reported that they noticed social exclusion (92%), followed by verbal aggression (37%), and instrumental aggression (30%). Similarly, Kwak and Kim (2016) reported that 63% of teachers in both kindergarten and daycare centers observed *hakkyo-pokryuk* among young children, and considered it to be serious and to warrant specialized education for its management.

However, few studies have directly questioned young children about bullying or victimization experiences. Lee et al. (2016) found that 8–13% of four- to six-year-olds participating in their study reported victimization experiences. Although children and teachers have reported bullying or victimization among young children, studies on the subject are still very few. Furthermore, studies of bullying in early childhood focused on teachers or children, rather than parents. There have hardly been any studies on how parents perceive bullying among young children in South Korea. Generally, mothers seem to be more sensitive to their children's bullying experiences compared to fathers, and they offer an effective solution to eradicate bullying (Georgiou 2008; Lester et al. 2017). Mothers in South Korea are more likely to be closely involved in the care and education of their children, compared to fathers (Hong and Lee 2019). Therefore analyzing mothers' thoughts on bullying among children will be useful in examining bullying in South Korea.

This study aimed to investigate mothers' perceptions of bullying among young children in South Korea, focusing on whether their children had experienced bullying, and their thoughts on these situations.

### 1.2. Early Childhood Education System in South Korea

In South Korea, children aged five years and below are educated in two types of educational institutions: kindergarten, where children aged three to five years are enrolled, and daycare centers, where zero to five-year-olds are cared for and educated. Parents can choose either of the two types of institutions. Most children (90%) aged three to five are educated/cared for at either kindergartens or daycare centers (Park et al. 2015). Children between the ages of six and eleven are enrolled in elementary schools, and the attendance rate of elementary schools in South Korea is 97.3% (Department of Education, Korean Educational Development Institute 2017). In keeping with government instructions, these institutions watch over children from roughly 9:30 a.m. to 1:30 p.m., working additional hours if necessary. If parents are unable to pick their children up in the afternoon, these institutions take care of the children until 5 p.m. or longer. Thus, young children spend most of their waking hours in one place with the same peers.

## 2. Method

This study used qualitative methods to explore the perspectives and experiences of bullying among mothers of young children.

### 2.1. Recruitment and Consent

Participants were recruited through snowball sampling. Mothers who were already familiar with the authors were interviewed first, and then asked to introduce other mothers with similar experiences. This study adhered to all ethical guidelines of the institution to which authors belonged to. There was no need for IRB approval for this research, as it does not involve any medical action nor cause harm. Before the interview began, the participants were informed of the research purpose, procedures, and data collection methods. A list of interview questions was also shown to them before the interview, so they were able to make an informed decision regarding their participation. Mothers who agreed to be interviewed also filled and submitted a consent form. Most participants were openly receptive and interested in this study, because they hardly had opportunities to talk about the issue of bullying or victimization among their children. Measures were taken to ensure that the collected data did not expose personal information such as the names of mothers and their children's educational institutions and areas of residence. The children's names quoted in the results section were indicated as alphabet initials (i.e., *A*, *B*, *C* and so on) with the exception of the child labelled M11, who divulged a verbal bullying incident that called for an indication of the child's name: in this case, a child's pseudonym was used. Further, participants were indicated as initials (i.e., M1: Mother 1, M2: Mother 2 . . . ).

### 2.2. Participants

Twelve mothers living in Seoul or Gyeonggi Province in South Korea, participated in this study. The participants were aged between 35 and 44 years, and were all married. Among the participants, six were not employed, three were part-time workers, two were full-time workers, and one was a postgraduate student. They had one to three children, and at least one of their children was between three and eight years old. The mothers were all college or university graduates. The information of the participants is described in Table 1. This study regarded early childhood as a development stage: from birth to eight years old (OECD 2015), in which children in kindergarten, day care center, and lower elementary school level were included. These children were exposed to bullying, but their bullying experiences were hardly investigated. Additionally, mothers whose children had joined elementary school recently were expected to give relatively richer information on their experiences related to bullying both in kindergarten and elementary school.

**Table 1.** Information of participants and their children.

| Mother | Number of Children | Child's Age (year) | Child's Gender | Child's Educational Institution Type |
|--------|--------------------|--------------------|----------------|--------------------------------------|
| M1 | 2 | 1<br>4 | Male<br>Female | No school<br>Daycare |
| M2 | 2 | 3<br>6 | Female<br>Female | Daycare<br>Elementary |
| M3 | 2 | 5<br>10 | Female<br>Male | Elementary<br>Daycare |
| M4 | 1 | 5 | Female | Kindergarten |
| M5 | 2 | 5<br>7 | Female<br>Female | Kindergarten<br>Elementary |
| M6 | 2 | 5<br>7 | Female<br>Female | Kindergarten<br>Elementary |
| M7 | 2 | 3<br>5 | Male<br>Female | Daycare<br>Daycare |
| M8 | 2 | 4<br>4 | Female<br>Female | Daycare<br>Daycare |

**Table 1.** *Cont.*

| Mother | Number of Children | Child's Age (year) | Child's Gender | Child's Educational Institution Type |
|---|---|---|---|---|
| M9 | 2 | 0.5 | Male | No School |
| | | 5 | Female | Kindergarten |
| M10 | 1 | 6 | Female | Elementary |
| M11 | 3 | 3 | Female | Daycare |
| | | 5 | Female | Daycare |
| | | 7 | Female | Elementary |
| M12 | 1 | 7 | Female | Elementary |

## 2.3. Interviews

Semi-structured interviews were conducted. All interviews were conducted by the first author of this study. Prior to the in-depth interview, preliminary interviews were administered to two mothers of four-year-old children to check the appropriateness of the interview questions and to confirm whether mothers of young children had experiences relevant to the research topic. Parents clearly indicated that their young children often experienced bullying from their peers or friends, and reported difficulties related to the incidents. They tried to solve them, but they did not know how to go about it. The preliminary interview showed that the mothers were aware of bullying and its specific features in early childhood, and that it was different from bullying experienced in other ages. The two mothers who were interviewed preliminarily were not included as participants in the main interviews. The main interviews were carried out either individually, or through a focus group, depending on the participants' schedule or their familiarity with each other. Seven individual interviews (M1, M2, M3, M7, M8, M9, M10), and two focus groups (Group 1: M4, M5, M6; Group2: M11, M12) were conducted.

Given the fact that several terms used in South Korea corresponded to bullying in western cultures, and that each term represented a slightly different meaning, the researchers did not use South Korean specific terms such as *wang-ta* or *goerophim* in the initial interview, in order to avoid the prejudice or preconception that mothers might have about the terms. These terms were used only after they were mentioned by the participants. Instead, cartoons describing several types of aggressive behaviors were shown to participants. Then, they were asked about their thoughts on them. The cartoons were developed by Smith et al. (2002), and they have been widely and effectively used to elicit the concept of bullying (Lee et al. 2011, 2012; Monks and Smith 2006; Smith et al. 2002). In this study, six cartoons, each describing different types of aggressive behaviors, were used: (1) an individual's physical aggression (hitting a smaller person), (2) verbal aggression (saying nasty things), (3) indirect physical aggression (breaking another's ruler), (4) group physical aggression (several children hitting a child), (5) direct/relational aggression (not allowing someone to play with others), and (6) indirect/relational (spreading a rumor) (A cartoon engaging in an individual's physical aggression is shown in Appendix A).

Mothers were shown the cartoons and asked,

- What do you think about these behaviors (expressed by the cartoons) among children?
- Has your child ever experienced (or have you ever heard about) these behaviors? If so, please tell me about the situation in detail.
- What did you do when your child had an experience related to these behaviors?

Interviews were carried out in cafes or the participants' homes, where the mothers felt most comfortable. The interviews lasted approximately 60–120 min, and were audio-recorded.

## 2.4. Qualitative Analysis

All interviews were transcribed into Korean. Nvivo 12 software was used to analyze the data, and the constant comparative method, based on the grounded theory, was used (Strauss and Corbin 1998). Basic concepts in open coding were identified. The categories and subcategories were named

by using words that represent and comprise the basic concepts. Some categories had subcategories as participants generated richer information and detailed perceptions on certain categories. The core categories were selected, and the relevance of the other categories based on elaboration were confirmed. Through this process, six major themes were identified and categorized. The authors independently conducted coding, created categories, and held discussions. Whenever there were differences in analysis, especially in categories or subcategories, the authors returned to the data, reviewed the analysis, and discussed it until they were in agreement.

## 3. Results

Six major themes emerged: (1) the concept of bullying (2) difficulty in defining bullying in early childhood (3) difficulty in communicating with other mothers about bullying (4) children who do not reveal their experiences (5) ways to be aware of bullying and (6) mothers' concern. Each theme consisted of categories that were sometimes divided into subcategories. Table 2 shows the hierarchy of themes and categories (and subcategories).

**Table 2.** Themes and categories (subcategories) that emerged from the analysis of mothers' interviews.

| Theme | Theme 1. Concept of bullying | Theme 2. Difficulty of defining bullying in early childhood | Theme 3. Difficulty in communicating with other mothers about bullying | Theme 4. Children who do not reveal their experiences | Theme 5. Ways to be aware of bullying | Theme 6. Mothers' concern |
|---|---|---|---|---|---|---|
| Category (Subcategory) | Categories 1-1. Criteria of bullying (Subcategories: Physical, psychological harm, repetition, power, intention, victim-centeredness | Category 2-1. Ambiguity of situation (Subcategories: Severity, repetition, self-defense, peer interaction) Category 2-2. Different standpoints of mothers (Subcategory: Aggressor's mothers vs. victim's mother, Mothers' personal characteristics) | Category 3-1. Relationship among others (Subcategories: Closeness, afraid of being *wang-ta*) Category 3-2. Links between mothers' and children's relationships | Category 4-1. Fear Category 4-2. Introversion, Category 4-3. Desire to play with aggressor Category 4-4. Limited language ability | Category 5-1. Online and offline network among mothers Category 5-2. Witness Category 5-3. Children's reports | Category 6-1. Worries of being bullies or victims Category 6-2. Consistency Category 6-3. Lack of coping strategies Category 6-4. Trust/distrust of teacher |

Due to the limited space, only the quotes that were most representative of each category (or subcategory) were shown. In some categories, more than one quote were displayed, while in other categories, only one quote was indicated if it carried the meaning of the category (or subcategory) efficiently.

**Theme 1. Concept of bullying.**

The theme concept of bullying explains how mothers conceptualized bullying, and what criteria they used to define bullying among young children. This was further explained by the category "criteria of bullying" and its subcategories.

*Category 1-1. Criteria of bullying. (Subcategories: Physical/psychological harm, repetition, power, intention, and victim-centeredness).*

Mothers' concepts of bullying were similar to those defined by academic researchers: They used the criteria of physical/psychological harm, repetition, power, and intention. Additionally, mothers focused on defining young children's bullying from victims' perspectives.

"Of course, hitting is *goerophim*, but psychological suffering also often occurs among girls" (M4). (Subcategory: physical/psychological harm).

"If this goes on for long, it is *goerophim*" (M2). (Subcategory: repetition).

"A child can once in a while engage in these behaviors for fun, but if those are repeated, they can be harmful" (M3). (Subcategory: repetition).

"The child (aggressor) does not consider his/her friend an equal" (M5). (Subcategory: power).

"If they know that they commit *goerophim*, then it is *goerophim*: even if they do not know what they do. If someone is victimized, that is *goerophim* regardless of whether the children are aware of the meaning of their actions" (M2). (Subcategory: intention).

"If the child (victim) is so distressed that he/she cannot sleep, then their experience is *goerophim*" (M6). (Subcategory: victim-centeredness).

"If an individual feels pain from an action directed at them, that is *goerophim*" (M4). (Subcategory: victim-centeredness).

Mothers often mentioned that stress or pain from the victims' perspective was an important criterion for defining bullying, rather than the use of relatively more objective standards (repetition, intention, power).

The subcategory "victim-centeredness" may have resulted from the ambiguity of the bullying situation in early childhood, which leads to difficulties in defining bullying in early childhood.

**Theme 2. Difficulty in defining bullying in early childhood.**

The theme difficulty in defining bullying in early childhood leads to obstacles that interfere in judging certain aggressive episodes as bullying or not. It consisted of the categories "ambiguity of situation," and "different standpoints of mothers", and subcategories of each category.

*Category 2-1. Ambiguity of situation. (Subcategories: Severity, repetition, self-defense, and peer interaction).*

Mothers have difficulty in defining bullying, although they are aware of the basic criteria (harmfulness, repetition, power, intention, and victim-centeredness), because of the ambiguity of the aggressive situation. This was further explained by subcategories such as severity, repetition, self-defense, and peer interaction. When aggressive behaviors occurred, it was difficult to decide to what extent these subcategories were involved (whether the aggressive behavior was severe, how many times it was repeated, whether it was done in self-defense, and whether it was negative peer interactions or general peer conflicts rather than bullying).

"*X* says nasty words and hits others. Are these behaviors that have resulted from having experienced humiliation? It is not clear whether *X* hit *Y* though *Y* has done nothing to him/her ... or if *X* swears at *Y*, and Y hits *X*, which behavior is more punishable?" (M5). (Subcategory: severity, self-defense).

"A teacher told children not to report an aggressive behavior until it had happened seven times. This is the solution provided by a teacher. The teacher said there was nothing wrong with the accounts of either of the children (the aggressor and the victim). Each had reasonable explanation for their behavior. Children only consider their own views. There is a gap between the children, which the teacher may have difficulty balancing ... It may be strange from an adult's perspective, but children feel victimized" (M1). (Subcategory: peer interaction).

Mothers and teachers were all confused by the boundary between bullying and other negative interactions (such as fighting and joking). Mothers were confused about what behavior was worse among several types of aggressive behaviors. Even more confusing was aggressive behaviors that occurred in response to others' provocation. Mothers stated that teachers regarded the aggressive behaviors among children as general conflicts rather than as bullying. It seems to be the reason why teachers did not react to every single conflict among children, and told children to report only when an aggression was repeated many times (such as seven times in the example above). In contrast, children were clear about these concepts: if they were at the receiving aggressive behavior, it was regarded as being bullied or victimized whereas if they were the aggressors, it was just for fun. Teachers' views were different from those of mothers, which caused mothers to distrust teachers, and explained the category of trust/distrust of the teacher under the sixth theme, mothers' concern.

*Category 2-2. Different standpoints of mothers. (Subcategory: Aggressors' mothers versus victims' mothers, and mothers' personal characteristics).*

Mothers had different perspectives, depending on whether their children were the victims or the aggressors. Their responses also depended on their personal characteristics. Some mothers, when their children were involved in bullying behavior, apologized to the victim's mother, which was regarded as reasonable and understandable by other mothers. However, some mothers of aggressive children reacted aggressively toward the victim's mother and did not admit their children's wrongdoing. Participants perceived that these different responses resulted from the mothers' personal characteristics. These could make them reluctant to tell other mothers about their children's victimization.

"My girl could not sleep and did not want to go to kindergarten. However, because it was not her girl who was victimized, she perceived the situation as simple and downplayed it, and that hurt me . . . They said my girl is too sensitive and fussy" (M6). (Subcategory: aggressors' mothers versus victims' mothers).

Mothers reported that aggressors' mothers tended to regard the incident not as bullying but a misunderstanding among children.

"They (aggressors' mothers) are not humble enough to apologize to victims or their mothers. They think there is something wrong with the other child (victim), or that there has simply been a misunderstanding between children" (M5). (Subcategory: aggressors' mothers versus victims' mothers).

"I told her frankly that I felt bad about her girl's careless words. *H* cries whenever *H* is reminded of the incident. It was hurting *H* . . . I know she was not at fault, her girl was . . . but she didn't take this situation as seriously as I did" (M8). (Subcategory: aggressors' mothers versus victims' mothers).

There were a lot of comments regarding these different points of view. Not all aggressive mothers justified their children's mistakes. Mothers' personal characteristics played a part in how they reacted upon hearing that their children were bullies. This is important, because their own nature makes the victims' mothers unlikely to tell aggressors' mothers about their children's bullying, which in turn, makes it difficult to stop such incidents.

"The first type, mother says, "No matter what, hitting is wrong and I apologize." This is normal. The second type says, "Oh, I will apologize . . . but maybe there is some issue among the children." This is still a mild reaction; the third is pushy, "No, there must be something wrong" and, ask her child. The child could be lying, but the mother chooses to only believe her child's words. This is what poses a problem" (M5). (Subcategory: mothers' personal characteristics).

**Theme 3. Difficulty in communicating with other mothers about bullying.**

The theme difficulty in communicating with other mothers about bullying represents mothers' reluctance and worries of telling or sharing bullying-related experiences of their children. The category "relationships among mothers" was deeply involved with the difficulties in communicating. Further, the category "links between mothers" and "children's relationships" explained that the relationships among mothers and those among children were connected and influenced each other, which made mothers more careful about talking about their or other children's experiences of bullying.

*Category 3-1. Relationships among mothers. (Subcategories: Closeness, fear of being wang-ta).*

Mothers found it difficult to directly talk to the aggressor's mother or to other mothers uninvolved in the bullying situation. They were afraid of damaging of their relationships with other mothers after telling experiences of their children.

The subcategory "closeness among mothers" can either make it easier or more difficult to tell other mothers of their children's bullying. In one way, closeness can easily solve this problem.

"If I know the aggressor and the aggressor's mother, I can approach her more easily, such as, 'Hey, your girl hits my girl, could you ask your girl about this?' However, if I do not know the mother, I would already be upset before asking her and would be ready for an argument" (M12). (Subcategory: closeness).

However, mothers of victims could also be unlikely to speak to aggressors' mothers about the situation if the two share a close relationship.

"She was not at fault, but we may be uncomfortable because of the children's matter. I pretended and tried to be fine because no one was (physically) harmed, but I am not actually . . . (fine)" (M6). (Subcategory: closeness).

Victims' mothers were unable to gauge how aggressors' mothers would react upon hearing of their children's behavior. Unless the aggressor's mother was sensible and empathized with the victimized child, the friendship between the mothers would end, or at least become uncomfortable.

The subcategory "fear of being *wang-ta*" explains that mothers could not tell victimization experiences of their children because they were afraid of being isolated if the victimization did not elicit other mothers' empathy. Mothers were worried about how the victimization of their child would be perceived, and whether other mothers could understand the situation.

"If I react very strongly even though it was the other child's fault, they might think that I overreact and regard me as a violent and stubborn person. I would become a cause for alarm'" (M5). (Subcategory: fear of being *wang-ta*).

Another mother's interview clearly demonstrated this worry.

"I have lived in this town for eight years, so I know the aggressor who makes trouble here . . . One child was victimized by this aggressor. The victim's mother is not one to remain silent; she made sure the whole school knew, sending instant messages directly even to her child's teacher. She opened a chat room for mothers, so that they could be aware of what happened among the children" (M2). (Subcategory: fear of being *wang-ta*).

M2 did not show a positive attitude toward the victim's mother; she seemed to think the situation could have been handled in other ways.

Generally, mothers know both aggressor and victim, because they live in the same town, and their children usually go to the same school in the district. Thus, their relationships are complicated: sometimes they may be close to aggressor's mother, in that case, they might tend not to blame aggressor too much.

*Category 3-2. Links between mothers' and children's relationships.*

Mothers' and children's relationships are related. Children's peer relations can be influenced by mothers' relationships. If mothers had conflicts, their children are not allowed to play together by their mothers.

"It is very common. Mothers' relationships directly affect those of the children. Children can recover their relationship, but they cannot play anymore because their mothers' relationship has been severed" (M7).

Sometimes, mothers can contribute to making a child or a child's mother *wang-ta* intentionally or unintentionally; they can stigmatize a child who has trouble with other children in the town, and the child and his/her mother can be socially excluded.

"If some mothers say that a child is weird, he/she is socially excluded, rumors spread implicitly around mothers; they overreact even over very trivial things . . . Mothers only think about their own children." (M9).

Mothers' selfishness in thinking only of their own children can be detrimental to the reputations of other children, contributing to making certain children *wang-ta*. In contrast, when an aggressive child bullied other children and his/her mother ignored the situation, other mothers were distressed by the mother and the child.

"If a child (aggressor) lacks self-control. He/she fights with others, then his/her mother should say 'it is your fault, you should have behaved better' but she responds, 'Um . . . I see' and that is the end of it. Then the boy would think 'Whatever I do is okay' Then, the mother gets in trouble with the other mothers, and she is socially excluded by the other mothers" (M6).

**Theme 4. Children who do not reveal their experiences.**

The fourth theme, children who do not reveal their experiences to, explained why children were unlikely to tell their bullying-related experiences to parents. The categories "fear", "introversion," and "desire to play with aggressor" were related to the psychological reasons for the children's difficulty in determining victimization. "limited language ability" was regarded as a cognitive factor that contributed to children's difficulty in reporting incidences of bullying to their parents.

*Category 4-1. Fear.*

If children were threatened or dominated by someone who is physically or socially more powerful, they felt fear, which made them difficult to report the bullying.

"The boys drank soap bubbles because they were frightened of the aggressor. There were 10 boys in the class, and five of them drank the bubbles" (M2).

"*X* commanded *A* to put up her hands and reflect upon her faults. I was speechless . . . The teacher told *X* that they must not do it" (M1).

The order of a dominant child seems to be undeniable in the children's social world. Children seem to panic, and are unable to think about what to do and how to cope with the dominant child's behavior.

*Category 4-2. Introversion.*

Some mothers worried that their children's introverted characteristics led to their being unaware of events in kindergarten.

"*X* is not weak nor tender like my girl . . . *A* is not talkative; usually she doesn't speak about what happens at the daycare center . . . She was so tender, she couldn't react to it" (M1).

Also, when a dominant child meets one who is tender and introverted, bullying is more likely to occur, as the introverted child is unlikely to directly express their displeasure to the aggressor.

"*O* is a dominant child, if there are children who are younger than *O*, *O* snatches what they are doing, and uses it" (M10).

*Category 4-3. Desire to play with the aggressor.*

Quite often, children seemed to think that if they reported the situation, they would be unable to continue playing with the aggressor.

This makes mothers frustrated.

"I told *E* not to play with the other girl, but she wants to, and I can't stop her, which really disappoints me" (M5).

Children seemed to want to play with aggressor, because they attracted characteristics that the aggressor had

"I asked *B* 'Why do you want to play with her?' and *B* said. 'Because she dresses like a princess and other children like her' . . . " (M2).

*Category 4-4. Limited language ability.*

Children have difficulty in expressing their victimization experiences, because their language ability restricts them from expressing their victimization experiences appropriately and in a timely manner. This was very frequently mentioned by mothers.

"*C* is a child of my friend. *C* has been suffering from atopic eczema since she was a baby, when she joined first grade in elementary school, she told her mother that she had had a very difficult time in kindergarten. Other kids used to tease her because of her skin. At the time, she had told her mother that she did not want to go to kindergarten, but her mother did not know the reason for this and pushed her to go kindergarten. Her mother had regarded her complaint as general or infantile behavior. She could not express what she was going through at that time because she was too young but she is mature enough now . . . " (M3).

"*J* was not able to speak in detail because she was four years old. She said 'Mom, *O* is really bad" I asked 'Why?, then '*O* is nice when teacher is hanging around, but she does whatever she wants when the teacher disappears' I said to *J* 'Children are like that . . . you also do whatever you want' . . . I did not think of it as important" (M10).

"A child cannot refute point-by-point when another child says that he/she has done something that he/she has not. He/she just cries and says, 'I didn't, I didn't.' The other children just see him/her crying; they don't listen to what he/she actually did, and it turns into a bad situation" (M11).

Young children's expressions are likely to be simple as the above (e.g., 'she is bad' 'I don't want to go kindergarten'). This can cause mothers to tend to disregard the importance of their expressions, and other children in the same class are not able to understand the situation.

Also, children were often unaware of whether the bad events that they experienced were bullying. Although they felt bad, it seemed to be difficult for them to tell at the time, which could be a result of their limited language ability, and their desire to play with the aggressor.

This did not mean that they were fine and not hurt; but rather, that they had buried the hurt within themselves for a long time, telling adults much later. This reflects on the fact that children do not forget their victimization. Instead, it seems that they can tell of when their cognitive or language development has reached a level where they are able to explain their experience.

Furthermore, young children could express non-aggressive behaviors which they received from others as aggressive behaviors, due to their limited language abilities, which might have been caused by their cognitive developmental stage in interpreting others' intensions and behaviors.

"I intentionally teach my boy to improve his language skills because he should be able to express himself if he is victimized, but that backfired. One day, a teacher called me and said that my boy told her I hit him. I was shocked, 'Did I hit him?' . . . I just patted him to put him to bed . . . " (M7).

Children's simple and delayed expressions of their victimization, or wrong expression of non-victimization experiences renders mothers unlikely to identify bullying incidents correctly. Also, as the incidents described by their children might have occurred months or even years ago, it is difficult for them to discern exactly what happened.

**Theme 5. Ways to be aware of bullying.**

The fifth theme, ways to be aware of bullying, outlines diverse ways in which mothers were informed of bullying incidents: the categories "online and offline networks among mothers," "witness," and "children's reports" further explained the theme.

*Category 5-1. Online and offline networks among mothers.*

Mothers heard of bullying incidents among children through group chat rooms, or from other mothers who were close to them.

"I didn't know at first, but I have heard from other mothers . . . They talked seriously about what had happened, and I asked them whose child had done it" (M2).

"The mum heard from another mum whose child is in the same class as her son" (M4).

Online chats among mothers were useful for obtaining information related to the children, but also had negative effects. Some mothers reported the following disadvantages of group chat rooms: "Group chat, especially among the mothers of all the children in one class, causes many problems" (M11).

"I only use it for class-related notices." (M12).

In spite of the negative effects of an online network among mothers, it is not easy for mothers to be logged-out from the online chat room, because some important messages or notices related to the school class are announced online.

*Category 5-2. Witness.*

Sometimes mothers witnessed what happened among the children, which often made them surprised or angry.

"Z commands my girl to obey her when they play together, and my girl cannot command as Z does. This happens consistently and it makes me so upset." (M8).

*Category 5-3. Children's reports.*

Children told their mothers about their victimization experiences, sometimes on the day of the incident, and other times, a while later.

"Usually, *B* does not tell me what happened in school. However, after that incident (victimization), she said, 'Mum, I dislike going to school a bit,' so I was aware that something was wrong" (M2).

"*J* does not tell me in detail, but she tells me when she is bullied. When she washes her hair, she says it . . . Children are like that, suddenly they are reminded of previous happenings" (M10).

**Theme 6. Mothers' concern.**

The last theme, mothers' concern, showed mothers' worries related to bullying. This theme consisted of the categories "worries of being bullies or victims," "consistency," "lack of coping strategies," and "trust/distrust of teacher".

*Category 6-1. Worries of being bullies or victims.*

The mothers worried about whether their children could be bullies or victims.

"*B* can be an aggressor, not a victim, so I warn her when she says something bad. Sometimes she says nasty words. I tell her that she should not say bad things to friends" (M2).

*Category 6-2. Consistency.*

From early ages, bullying happened frequently, and mothers were afraid that it would be consistent throughout mid-childhood or adolescence.

"I thought these things don't happen at early ages. However, such incidents occur even among first and second graders in elementary school. I worry about what will happen when they grow up" (M6).

"They met when they were very young. It happened sometimes but disregarded at that time. It continued in elementary school, and now *S* slaps *T*'s face" (M3).

As M3 mentioned, mothers perceived that a failure to intervene in children's problematic peer relations from an early age could cause the behaviors to become severe.

*Category 6-3. Lack of coping strategies.*

Mothers were embarrassed and surprised when they heard that their children were being bullied. Generally, they did not know what to do. They told teachers, or asked their children how to react, but were unsure of whether their interventions were appropriate and effective.

"I was so embarrassed when I heard of it. I did not know what to do" (M4).

"I did not know what to do, and to what extent I could intervene in a child's matter . . . Even when I say something, I cannot blame or tell the boy off . . . he may say 'I just call *Oh-e* to *Oh-ri*,' boys are not afraid (of my words)" (M11).

The mothers' responses reflected that they did not have many options for coping. M11's daughter (*Oh-e*) was verbally bullied because her name had similar pronunciation to *Oh-ri* (in Korean this means "duck"). M11 expected that the aggressor would not listen to her words, as he knew that she could not speak assertively to him. Thus, he justified or trivialized his behavior, rather than admitted it as bullying.

*Category 6-4. Trust/distrust of teacher.*

Mothers expected that the teacher would be able to solve bullying incidents. Sometimes their expectations were fulfilled, but sometimes they were not. Depending on teachers' reactions or coping mechanisms, mothers built trustful/distrustful relationships with the teachers. Some mothers formed trustful relationships with teachers while experiencing and dealing with their children's victimization.

"Last year, the teacher was very helpful. The teacher told me to ask *A* if she had said bad things. The teacher told me to ask *A* if she had said bad things. The teacher said that children tend to disclose only specific parts of stories, where they appeared to be the victims. So, I needed to find out what exactly my child's role had been in the matter" (M1).

In contrast, some mothers were disappointed by teachers' passive responses to their children's experiences of victimization. Although they understood the teachers' stance, they were frustrated by the fact that the matter was not dealt with adequately.

"I told *Oh-e* to ask for help from the teacher. However, the teacher told children not to complain, but to try solving their issues independently. The teacher considered children's reporting as tattling. The teacher has difficulty with looking into every report because children complain about even the most trivial things on a daily basis. The boundary between tattling and reporting is not clear. "I do not know to what extent I have to tell *Oh-e* 'you have to report to the teacher if bad things happen to you.' She was hurt, cried, and felt she was being bullied and that no one could help . . . " (M11).

"I talked through the phone with teacher, the teacher did not consider seriously . . . The teacher just seemed to think 'Um . . . another mother phones me'" (M10).

As mentioned in the subcategory "peer interaction" under *category 2-1. Ambiguity of situation*, teachers and mothers have different views. Mothers desired the teachers' help in their children's victimization; however, from the mothers' viewpoints, teachers were perceived to handle general conflicts and victimization in the same way, even though they should be treated differently.

## 4. Discussion

This study showed that bullying among young children occurred explicitly, and in a very sophisticated way. The difficulty in defining bullying among young children arises from the various perspectives on the matter, which hinders communication between people, eventually making the situation difficult to solve.

### 4.1. Mothers' Definition of Bullying among Young Children

Mothers defined the phenomenon of bullying in early childhood as being similar to bullying in middle childhood and adolescence, including the concepts of aggressive behavior, repetition, and an imbalance of power (Olweus 1993). However, one more concept—victim centeredness—was emphasized in the context of early childhood bullying.

The victim-centered aspect of defining bullying is related to the issues of repetition, harm, and the severity of aggression. One physical attack can be categorized as bullying if the child on the receiving end is frightened and damaged by it (Arora 1996), and it does not necessarily need to last a long time. According to Rigby, saying "You weren't bullied. You will never meet the guy again" (Rigby 2006, 32) is not practical. Furthermore, young children's language ability, and the relationships among mothers emphasize the need for victims' perspectives on defining bullying. Due to their limited language ability, children's bullying or victimization situations were less likely to be described accurately. Also, mothers had different perceptions on bullying, depending on whether their children were bullies or victims. These are specified in detail in the following section.

### 4.2. Relationship among Mothers and Its Link to Children's Relationships

Mothers had difficulty in talking about or reporting bullying/victimization incidents, because of their own relationships. Mothers had different perspectives toward bullies and victims, and their personal characteristics contributed to this dichotomy. Their difficulty in ascertaining whether a certain behavior was counted as "bullying" or "general conflict", was consistent in previous studies (Mishna et al. 2006; Purcell 2012). This study provides further reasons for the diverse perspectives on bullying.

Different perspectives toward bullying resulted in anxieties that other mothers, who might judge the situation differently, may not understand or empathize with them. They were afraid of being excluded from mothers' social circles.

Mothers' and children's social exclusion or bullying were interrelated, and they influenced each other. This was an interesting finding in this study. On one hand, a child's bullying can result in a mother's social exclusion. If the aggressor's mother does not admit that her child's behavior constitutes bullying and instead blames the victim, other mothers may avoid or socially exclude her. On the other hand, a victimized child can cause his/her mother to be excluded. If the victim's mother strongly puts forth her opinions on the victimization of her child, other mothers might think she has gone too far. This is in the similar line with a previous finding; parents were reluctant to be open about the victimization of their child, because they were afraid of being labeled as overprotective, which could influence their child's reputation in kindergarten (Crisp and Humphrey 2008).

Regardless of whether their children are aggressors or victims, mothers can be isolated if they do not accommodate others' perspectives, and are adamant about their own stance. Mothers' group chats can contribute to children's and mothers' bullying. These group chats are not only useful for obtaining information related to children and school work, and for communicating with other mothers, but they make it easy to manipulate relationships, or to spread rumors about happenings among children. Owing to this reason, some kindergartens and schools in South Korea discourage mothers' group chats. In this way, early childhood bullying can be connected to cyberbullying and bullying among mothers.

This implies that it is imperative for mothers to be educated about cyber and traditional bullying. Mothers in group chats could intentionally or unintentionally reveal information that is related to other children, and the other participants of the chat may then become prejudiced toward the children and their mothers.

Mothers were in a dilemma over whether or not to tell others about bullying incidents. If they revealed the incident, they were afraid of being misunderstood by other mothers, and if they did not, their children could be victimized again.

Therefore, mothers generally tried to solve the problem themselves by teaching their children coping strategies, or speaking to the teachers. Sometimes they told the mother of the aggressor, but this was regarded as a risky move, because it could either solve the problem or make it worse.

### 4.3. Children's Difficulties

Children can have difficulty in expressing their victimization experiences, because of their psychological traits (introversion) emotional status (fear), desire to play with the aggressor, and limited language abilities.

Introverted or shy children may be distressed by extroverted or dominant children who strongly express themselves. Socially excluded girls seemed to try not to ruin other children's moods, and to fit in, rather than dwelling on how being excluded made them feel. They might have been under the impression that their relationships will end if they express their negative feelings. Although it was not reported whether the victimized children blamed themselves or not, this may be risky because it allows bullying to continue, negatively affecting the victim's mental health by lowering their self-worth.

Furthermore, the aggressor does not always play that role: he/she could also be a cheerful mate during play. Due to the desire to keep the aggressor as a playmate, children might overlook their victimization experiences in spite of their unpleasant feelings. This does not mean that they forget their victimization: these experiences remain in their memory for a long time, which reflects that the experiences were hurtful for their emotional and mental health. They just might not have known how to express themselves, or how to deal with the unpleasant feelings at the time. This implies that young children must be taught coping strategies and emotion management skills.

Most mothers reported being surprised at the occurrence of bullying at young ages, and worried about its persistence until adolescence. Mothers reported that they did not know what to do when they heard of the victimization of their children. They expected teachers' active involvement in the issues, but this was not always the case. This is the same line with previous findings in that parents were upset and surprised by their children's victimization, and upon seeking help from the school, they were disappointed by the school's responses and felt helplessness about the situation (Brown 2010).

### 4.4. Suggestions for Interventions in Early Childhood Bullying

Given the findings in this study, it is necessary to develop prevention/intervention programs for bullying in early childhood. While at the school level, there are strict and explicit instructions for what to do in the event of bullying, no such instructions exist in educational institutions for young children. Intervention programs should include the following aspects.

First, education on the concept of bullying should be implemented for mothers, teachers, and children. Helping them to conceptualize the behaviors that can constitute bullying could help to fill the gaps in their perspectives. Then, it can be possible for mothers to discuss the wrong behaviors of children without damaging their relationships. It would also be helpful for teachers to understand that bullying can be defined differently among parents, children, and themselves, and to try to empathize with mothers' and children's perspectives. Although defining bullying is complicated, this procedure is necessary for recognizing bullying, and intervening (Sawyer et al. 2011). Parents' involvement (i.e., information for parents, parents' training) in intervention was found to be an important component in anti-bullying programs (Ttofi and Farrington 2011). However, there are very few studies on education for parents on the concept of bullying in early childhood. This should be considered while developing an intervention program for bullying among young children.

Second, it is important for children to learn to express their experiences in an appropriate way. Young children may have difficulty objectively describing certain situations or experiences, because of their language development levels, as well as self-centric thoughts. Adults can support them in expressing their needs and expectations in appropriate ways. It is essential to educate young children to understand others' intentions.

To define bullying from victims' perspectives (victim-centeredness) appropriate expressions related to their bullying or victimization experiences is necessary. At the same time, mothers and teachers need to be aware that even one incident can count as bullying for young children, whether the incident is severe or not. Adults need to be sensitive and pay attention to what children say and how they behave. Adults may dismiss what children say to them, because they may often regard children's words as being neither severe nor important (e.g., a child kept saying "I don't want to go to kindergarten" but her mother ignored it). However, young children's perceptions of the severity of the incident may differ from that of adults.

It is also imperative to teach children coping strategies: bullied children must learn how to react, and express and report their victimization; and aggressors must learn about how others may feel regarding their behavior, and ask for their needs in a socially appropriate way. Social and emotional skills have been included in intervention programs for young children. For example, 'Second step: A bullying prevention program' focused on developing social skills among children who are four to eight years old. As preventive strategies, controlling impulsive behavior, anger management, and perspective-taking skills were practiced by children (Committee for Children 2014). The program improved children's social emotional competence, and reduced peer problems (Low et al. 2015).

Third, social efforts are necessary to establish a sound cyber culture among mothers. This has not been elucidated in any intervention program; however, given the findings in this study, this should be included in intervention programs for families or parents. Education on cyber *goerophim* (cyberbullying) for mothers should be implemented. It can prevent connecting bullying among children to bullying or cyberbullying among mothers, thereby improving mothers' sensitivity toward cyberbullying. The power of mothers' cyber communities can be used in a positive way. They can be bystanders who pay attention to the incidents that occur among mothers or children, and judge whether certain behaviors can be regarded as bullying or not. They could encourage the mothers of victims and protect the children whose rumors are being spread.

This study has limitations. First, it investigated mothers' perceptions only, and not those of fathers, and most mothers in this study were relatively highly educated. Parents may show different perceptions on bullying. For example, fathers may be more likely to think that bullying can happen

while children grow up. Thus, one need to be careful when generalizing the results of this study. Perceptions of mothers who come from more diverse demographic backgrounds are needed.

Second, teachers' voices were not investigated in this study. According to mothers' reports, teachers seem to have more generous views on bullying. Further study is necessary about how teachers' perspectives differ from those of mothers.

Third, it is not known whether relationships among mothers are important in understanding the phenomenon of young children's bullying in other cultures. The replication of this study in other cultures would be helpful in comparing the similarities and differences of mothers' (or parents') perceptions of bullying across cultures.

Last, the expectations from the interventions should be studied further. This study analyzed only mothers' responses to their children's bullying experiences, rather than the need for intervention programs.

## 5. Conclusions

This study shows that understanding bullying among young children is quite complicated. The ambiguity involved in defining bullying, the diverse perspectives among mothers on bullying, the relationships among mothers, and the relationships between mothers and children lead to both mothers' and children's difficulties in speaking of and sharing the victimization experience. Sharing one's victimization experience is important in solving the problem by empathizing and understanding each other. This does not show that bullying among young children is not severe; rather, its severity has not been disclosed nor shared among people. The South Korean government and society must pay attention to bullying among young children. It may contribute to emphasizing parental involvement in bullying studies, though this study was conducted by using a South Korean sample.

Finally, this study can contribute to understanding the nature of bullying in early childhood, improve the attention paid to bullying among young children, and emphasize the need for intervention/prevention programs.

**Author Contributions:** Conceptualization, H.-j.J.; Data curation, H.-j.J. and S.-h.L.; Formal analysis, H.-j.J. and S.-h.L.; Funding acquisition, S.-h.L.; Investigation: H.-j.J.; Methodology, S.-h.L.; Resources, S.-h.L.; Writing original draft: S.-h.L.; Writing—review & editing: H.-j.J. and S.-h.L.

**Funding:** This research was supported by the Chung-Ang University Research Grants in 2018.

**Acknowledgments:** We deeply appreciated the mothers who were willing to participate in, and shared their experiences for this study.

**Conflicts of Interest:** The authors declare no conflict of interest. The funders had no role in the design of the study; in the collection, analyses, or interpretation of data; in the writing of the manuscript, or in the decision to publish the results.

## Appendix A

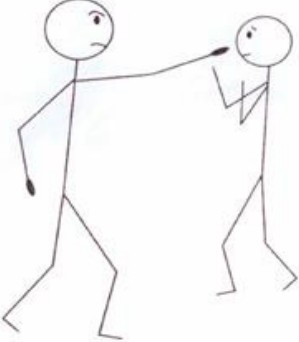

**Figure A1.** A cartoon: Individual physical aggression.

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
