# Peer review of "Mothers’ Perceptions of the Phenomenon of Bullying among Young Children in South Korea"

_socsci, doi:10.3390/socsci8010012_

Round 1

Reviewer 1 Report

This manuscript, “Mothers’ perceptions of the phenomenon of bullying 2 among young children in South Korea,” investigated South Korean mothers’ perspectives on bullying in early childhood and identified six common themes using Nvivo program. This was a well-written, easy to follow study and investigated an important topic relevant to the readers of Social Sciences. This manuscript has a number of strengths and contributes to the literature on bullying among South Korean young children. There are, however, a number of issues that need to be addressed. I hope my recommendations will be helpful to the authors as they work to revise this manuscript.

Abstract

-The abstract talks about different views of bullying across mothers, teachers, and children, but authors only conducted interviews with mothers, and therefore, the gathered information does not reflect teachers’ and children’s perspectives on bullying and it’s hard to say their views are different.

Introduction

-Introduction has a thorough review of literature on bullying from different cultures as well as specific to South Korea.

Method and Results

-Please provide information regarding IRB approval

-Please describe the interview process more in details. Was this one-on-one interview or two interviews with one or more mothers? How did authors make sure to provide consistent interview experiences across different interviewers and mothers?

-What was the rationale or criteria for authors to select and include the themes and categories? For example, I noticed that for some categories, multiple mothers contributed to the category by providing their perspectives on the topic. However, for other categories, only one (e.g., Category 5-2) or two (e.g., Category 4-3) contributed, and I am not sure if having one or two mothers expressed their perspectives on the matter is sufficient enough to create a category on the topic.

Discussion

-I would like to see more implications based on this study’s findings. For example, as authors mentioned, this additional concept – victim centeredness – was emphasized in the context of early childhood bullying. What would be the best way to incorporate this concept into developing prevention and intervention efforts for teachers, parents, and students?

Overall, I enjoyed reading this manuscript and learning about some similarities and differences in mothers’ perceptions of bullying happening with their young children between South Korean culture and other more frequently studied cultures in previous literature. However, there are many typos and grammatical errors noticed. For example, on page 1, the last sentence “three-year-old children(should be a space between these two words)and in under” and on page 2 “Therefore, adults’ awareness of accurate concept of bullying is” is an incomplete sentence. Please review the manuscript thoroughly to remove those errors.

Author Response

The file of response to reviewer 1 was attached.

Reviewer 2 Report

This manuscript is a study of mothers' perception on bullying among young children in Korea. Many of the bullying studies in Korea are about school aged children. Since there is not enough information about bullying among young children, it is a meaningful study to lay the foundation for bullying-related research, using a qualitative research method. In order to become a better study, I kindly propose that the following be corrected:

[Introduction]

1. I wonder if there is a special reason why you have offered various terms for bullying that are being used in Korea. If you want to introduce these different terms, you need to explain enough about the differences. Currently, only a variety of terms are described, but it is not clear why they were introduced and how they relate to bullying. Please refine your terminology and clarify which terms you will use in this study. And isn't “gorophim” too different from the '괴롭힘' pronunciation? In my opinion, if you want to write Korean(Han-gul), “geau-rop-him” would be appropriate.

2. In line 91 from 2 of 14, a study by Kwak and Kim (2016) on daycare centers and kindergartens showed up to 63 percent of school violence. In Korea, school violence is a term that should be used at least elementary school and includes a wider range of behaviors than bullying. As mentioned above, there will be no confusion when the terms are well organized.

3. The main purpose of this study is to explore how mothers perceive bullying among young children. But the introduction only mentions that there is no research on it. It doesn't say why it matters that how mothers think of bullying is important. I want you to supplement theoretical background on why you studied mother's thoughts, not children’s, father's and teacher’s.

4. In addition, there was not enough evidence to explain why study on bullying among young children are needed. Rather than simply showing the percentage of bullying among young children, it would be better to add developmental importance, and theoretical background to show why you should pay attention to the bullying among young children.

[Method]

1. The participants were mothers of children from 3 to 8years old. Please explain why you have included these participants? Also, I am wondering whether it is Korean age or not. If so, it looks like children in the first grade of elementary school were included, too. The authors explained the children who are in daycare centers and kindergarten were participated. Please explain this discrepancy. And there are significant differences in peer relationships and cognitive abilities between 3years old and 8years old. Can children of these two ages be grouped together? Please provide detailed criteria for participation in the study.

2. Not enough information about Participants was provided. It would be helpful to understand if I could match each participant's child age and gender in detail. Especially, author mentioned that the current study is about young children who were in daycare center and kindergarten. However, it is very confusing because the interview material included event at the Elementary schools. Please suggest for detailed information about mother and their child.

3. The author mentioned that pilot study was conducted with two mothers. It is not clear whether they're among 12 people or not. Please describe what you found through your pilot study and how you applied findings to the current study.

4. The author used cartoons for an interview with mothers. I wonder why the author used cartoon for the current study. Also, specific information about the overall method should be provided, including who made cartoons, how many cartoons were used, and what kind of bullying was described in cartoons. It would be helpful, if cartoons provided in appendix.

[Results]

1. I'd like to recommend clearer organization of results. Especially, among the terms proposed in subcategory, it is difficult to relate them to the interview contents such as victim centered, mother's personal characteristics (rational? irrational?), and fear of being Wang-ta. Also, if you clarify why each subcategory was named, the study will be well understood.

2. Interview contents are described with considerable confusion. I'd like to encourage you to think about how you can deliver results more effectively.

3. The content of the interview mentions child's name. Is there a reason why you should mention it yourself? I'd like to recommend that this be anonymous because it can also be ethical issues.

4. In the interview, it appears that the authors are naming participating mothers as M1 and M2. If you want to use the abbreviation M, you must first demonstrate that M means mother. And through interview contents, I realized that most of them are focused on M2, M5, and M6's interview. The author need to explain why their interviews are especially mentioned, why other mothers are missing. In particular, I don't think the M10 interview is included, so I'd like to hear about it.

[Discussion]

1. In the children's difficulties section, the theoretical background must be complemented. Especially, the authors argued that child should learn strategies for dealing with bullying and how to regulate emotions. It would be more reliable, if the evidence that it actually works when children learn about it. Also, at the end of this section, the authors mentioned roles of teachers. It is hard to understand how this relates to children's difficulties.

2. The current study is a significance of this research when we understand the perception of bullying by mothers and how it will be used in intervention. The author describes Intervention relatively well but does not suggest concrete evidences. Please organize authors' ideas about how to educate parents, children, and teachers in daycare center, and kindergarten. If the authors provide evidences of existing prior research, the authors' argument would be reliable.

3. Finally, in the discussion, the author states that the Korean government should actively intervene bullying among young children. I think this research particularly valuable when published in the Korean journal for policymakers and educators who prepare bullying intervention. Nevertheless, I would be grateful if you could further explain why you submit this manuscript to the current journal (social science) which is for foreign readers.

I really appreciate your efforts to conduct challenging research to understand bullying among young children in South Korea.

Author Response

The file of response to the reviewer 2 comments was attached.

Round 2

Reviewer 2 Report

Dear. Authors

Thank you for you endeabors to improve manuscript.

It looks more clear, and organized than prior version of manuscript.

One thing what I want to suggest is that some research background about "Korean" bullying in Kindergarten. It will have more valuable information what you want to share with readers. 
 It is because in the discussion, you mentioned that it is important to share about bullying among Korean population. 

Although this study is kind of basic steps, it is great to understand bulliyng in kindergarten. I hope that this research would be background of next study which apply more scientific methods.

Thank you for your hard work.